# Metabolic Signatures of Tumor Responses to Doxorubicin Elucidated by Metabolic Profiling *in Ovo*

**DOI:** 10.3390/metabo10070268

**Published:** 2020-06-28

**Authors:** Iman W. Achkar, Sara Kader, Shaima S. Dib, Kulsoom Junejo, Salha Bujassoum Al-Bader, Shahina Hayat, Aditya M. Bhagwat, Xavier Rousset, Yan Wang, Jean Viallet, Karsten Suhre, Anna Halama

**Affiliations:** 1Department of Physiology and Biophysics, Weill Cornell Medicine—Qatar, Doha 24144, Qatar; iwa2004@qatar-med.cornell.edu (I.W.A.); sak2041@qatar-med.cornell.edu (S.K.); ssd2002@qatar-med.cornell.edu (S.S.D.); shh2026@qatar-med.cornell.edu (S.H.); kas2049@qatar-med.cornell.edu (K.S.); 2Breast Cancer Unit, Hamad General Hospital, Hamad Medical Corporation, Doha 3050, Qatar; KJunejo@hamad.qa; 3Department of Medical Oncology, National Center of Cancer Care and Research, Hamad Medical Corporation, Doha 3050, Qatar; Sbujassoum@HAMAD.QA; 4Scientific Service Group of Biomolecular Mass Spectrometry, Max Planck Institute for Heart and Lung Research, D-61231 Bad Nauheim, Germany; Aditya.Bhagwat@mpi-bn.mpg.de; 5INOVOTION SAS, 38700 La Tronche, France; xavier.rousset@inovotion.com (X.R.); yan.wang@inovotion.com (Y.W.); jean.viallet@inovotion.com (J.V.)

**Keywords:** chicken chorioallantoic membrane (CAM) system *in ovo* model, triple negative breast cancer, doxorubicin treatment, cancer survival mechanism, metabolomics, lipidomics

## Abstract

Background: Dysregulated cancer metabolism is associated with acquired resistance to chemotherapeutic treatment and contributes to the activation of cancer survival mechanisms. However, which metabolic pathways are activated following treatment often remains elusive. The combination of chicken embryo tumor models (*in ovo*) with metabolomics phenotyping could offer a robust platform for drug testing. Here, we assess the potential of this approach in the treatment of an *in ovo* triple negative breast cancer with doxorubicin. Methods: MB-MDA-231 cells were grafted *in ovo.* The resulting tumors were then treated with doxorubicin or dimethyl sulfoxide (DMSO) for six days. Tumors were collected and analyzed using a global untargeted metabolomics and comprehensive lipidomics. Results: We observed a significant suppression of tumor growth in the doxorubicin treated group. The metabolic profiles of doxorubicin and DMSO-treated tumors were clearly separated in a principle component analysis. Inhibition of glycolysis, nucleotide synthesis, and glycerophospholipid metabolism appear to be triggered by doxorubicin treatment, which could explain the observed suppressed tumor growth. In addition, metabolic cancer survival mechanisms could be supported by an acceleration of antioxidative pathways. Conclusions: Metabolomics in combination with *in ovo* tumor models provide a robust platform for drug testing to reveal tumor specific treatment targets such as the antioxidative tumor capacity.

## 1. Introduction

The molecular landscape of breast cancer gene expression defines its specific types, drives treatment strategies, and serves as a predictor of treatment response and overall prognosis [1]. Breast cancers lacking estrogen receptor (ER), progesterone receptor (PR), and human epidermal growth factor receptor 2 (HER2) overexpression/amplification are known as “triple negative breast cancer” (TNBC) [2]. The aggressive clinical nature of TNBC together with a lack of suitable therapeutic molecular targets are the main reasons leading to an overall poor patient prognosis [3]. The treatment strategies for TNBC are limited to chemotherapy, frequently consisting of a doxorubicin regimen combined with paclitaxel as a neoadjuvant or adjuvant setting to target the highly proliferative nature of TNBC [4]. Although the initial response to chemotherapeutic treatment is more favorable in patients with TNBC than in patients with other breast cancer types, the overall survival and outcome is worse due to a frequent disease reoccurrence and together with developing resistance, underscores the need for new targets and the development of more efficient treatment strategies [5].

Enhanced and uncontrolled cancer cell proliferation, a key feature of TNBC, was shown to be associated with alterations in cancer cell metabolism that support cancer cell growth and division [6]. Elevated aerobic glycolysis with simultaneous lactate production was initially observed as abnormal cancer metabolism back in 1925 [7]. Technical advancements in the field of mass spectrometry and the emergence of metabolomics today allow for the broad profiling of the small molecule (metabolite) composition, largely contributing to recent progress in the field of cancer metabolism [8]. These techniques showed that the metabolic characteristics of TNBC are associated with elevated glycolysis, oxidative phosphorylation, and choline metabolism, known as cholinic phenotype as well as increased levels of glutathione and glutamate together with decreased glutamine levels [9,10,11]. 

Glucose catabolism contributes to nucleotide, amino acid, and fatty acid synthesis [12], thus plays a pivotal role in the maintenance of high levels of glycolytic intermediates supporting anabolic reactions and biomass generation [13,14]. Similarly, glutamine is utilized as a source of biomass by TNBC [10,15], but also as a component contributing to the generation of cell antioxidative capacity via glutamate and glutathione synthesis [11]. The cholinic phenotype not only provides structural molecules for cellular membrane generation, but is also associated with oncogene signaling, malignant transformation, and tumor progression in TNBC [16].

Doxorubicin, a first-line chemotherapeutic drug used to treat TNBC patients [4], targets the highly proliferative feature of TNBC by inhibition of the enzyme topoisomerase II and intercalation within DNA base pairs, which results in breaks in the DNA strands and consequential inhibition of both DNA and RNA synthesis [4]. Free radicals causing damage to cellular membranes as well as to DNA and proteins are also generated under doxorubicin treatment [4]. However, rather than complete remission following doxorubicin regimens, frequently only partial responses are observed, followed by acquired resistance [5]. The limited responses of patients might suggest the activation of molecular and metabolic mechanisms that enable cancer cells to survive the initial treatment. In line with this hypothesis, cancer cell resistance to doxorubicin has been linked to ceramide metabolism and an upregulation of glucosylceramide synthase (GCS) gene expression [17,18]. Another study suggested a role of the accelerated glycolytic pathway in doxorubicin resistance, stimulated by the fibroblast growth factor receptor 4 (FGFR4) [19]. However, these studies are most often based on monitoring a single pathway, so alternative interpretations cannot be ruled out. A systemic view on doxorubicin-induced metabolic rewiring, which can be achieved by an untargeted metabolomics approach, could improve this situation and provide further insight into activated cancer cell survival mechanisms activated under the given treatment. 

Another major axis of research in the study of tumor resistance to chemotherapy is the development of affordable and easy-to-use realistic three-dimensional (3D) in vitro and in vivo models for individual patient tumors [20]. For instance, 3D culture of primary cells, known as organoids, were shown to closely resemble a heterogenetic tumor microenvironment including morphological, genetic, and epigenetic signatures of tumor [21]. The surgically removed tumor sections from patients transplanted into immunodeficient mice allowing for tumor growth from the individual patient, known as patient derived xenograft (PDX), were shown to reflect on cellular and histopathologic features of parent tumor [22]. Significant advancements in the context of cancer drug testing, aimed at facilitating personalized medicine, have been achieved by the implementation of both tumor derived organoids and PDX mice models [21,22]. However, metabolomics study in organoids might be challenging due to the sample matrix, and considering that PDX models have accompanying time- and cost-efficient constraints.

Alternatively, the chicken chorioallantoic membrane (CAM) (*in ovo*), considered as an in vivo experimental model, is highly suitable for tumor engraftment due to the membrane vascularization supporting tumor growth as well as the lack of a fully developed immune system in the chick embryo until day 18 of embryo development [23]. This model has been deployed to study cancer biology in multiple contexts including angiogenesis, metastasis, early toxicity studies, and anticancer drug studies [23,24,25,26,27], and could serve as a robust alternative to simple organoids and expensive PDX mice models. For instance, *in ovo* experiments can be conducted in a relatively short time-frame of 18 days. They are a low-cost model and the formed tumors are easily retrievable [28]. Moreover PDX *in ovo* has been recently suggested for robust drug testing, allowing for large-scale screening of pharmacologic agents [24]. 

However, to the best of our knowledge, metabolic profiling has not been applied to monitor anticancer drug responses *in ovo*. Such a strategy could serve as a versatile platform, offering a comprehensive view of relevant metabolic processes in the context of cancer resistance and treatment optimization. Here, we assess the feasibility of a metabolomics implementation for anticancer drug testing *in ovo.* As a proof-of-concept, we provide a comprehensive overview on metabolic processes driven by doxorubicin treatment in the tumor. The TNBC cell line, MB-MDA-231, was grafted on CAM and the tumors were treated with doxorubicin or vehicle (DMSO). We conducted global metabolic profiling using the Metabolons HD4 platform [29] and the Lipdyzer complex lipid platform (CLP) [30] on collected tumors to gain detailed insight into metabolic rewiring triggered by doxorubicin. Our study shows the feasibility of using metabolomics in *in ovo* models for the monitoring of anticancer drug responses. We also determine the metabolic pathways activated under doxorubicin treatment, which might be considered as a cancer survival mechanism and thus represent potential treatment targets. 

## 2. Results 

### 2.1. Doxorubicin Treatment Suppresses Tumor Growth in Ovo 

We monitored the impact of doxorubicin on triple negative breast cancer cell line MB-MDA-231 in vitro. The treatment with two different doxorubicin concentrations (1 µM and 5 µM), but not with DMSO, resulted in time- and dose-dependent decrease in the cell proliferation and viability (Figure 1A,B). The viability decreased to 20% and 60% in the cells treated with 1 µM and 5 µM of doxorubicin, respectively. 

Next, we investigated whether doxorubicin would impact tumor growth of MB-MDA-231 cell line grafted *in ovo*. The MB-MDA-231 cell line was propagated and the cells were grafted onto the CAM of the egg at day 9 of chick embryo development (Figure 1C). The eggs were randomly assigned into three groups including untreated (control) and treated with vehicle (DMSO) or doxorubicin. The tumors grew over a period of eight days. The treatment with DMSO and doxorubicin started at day 2 after the cancer cell grafting and was conducted every 48 h. The untreated tumors serve as a positive control. The tumors were collected at day 9 after the cancer cell grafting (Figure 1C). The tumors treated with doxorubicin were smaller in size in comparison with the vehicle and control group, and there were no differences between the control and vehicle treated tumors (Figure 1D). We further examined the tumors’ weight and found significant differences between the doxorubicin treated tumor mass and control as well as the vehicle treated tumors. There were no significant weight differences between the control and vehicle treated tumors (Figure 1E). The mass of doxorubicin treated tumor was 50% reduced in comparison with the vehicle and untreated group. These data demonstrate the significant impact of doxorubicin on cancer cell proliferation in vitro and *in ovo*. The tumor growth *in ovo* was suppressed, but not completely inhibited. 

### 2.2. Metabolic Profiling in Ovo Reveals Alterations in Tumor Metabolism Triggered by Doxorubicin Treatment 

Given that doxorubicin treatment *in ovo* resulted in 50% of tumor growth suppression, it could be hypothesized that cell population, which proliferated, resists the treatment by activation of cancer survival mechanisms. We further reasoned that those mechanisms could be revealed by the analysis of metabolic alterations triggered by doxorubicin treatment. To test this hypothesis, we conducted metabolome-wide profiling on the collected tumor samples using two mass-spectrometry based platforms including the broad metabolic profiling HD4 platform [29] and the Lipdyzer complex lipid platform (CLP) [30]. We identified 556 metabolites involved in eight primary metabolic pathways including amino acids, carbohydrates, cofactors and vitamins, energy, lipid, nucleotides peptides, and xenobiotic metabolism on the HD4 platform (Figure 2A) and 956 lipids predominantly triacylglycerols, phosphatidylethanolamine, sphingolipids and diacylglycerols on the CLP platform (Figure 2B). 

We conducted the principal component analysis (PCA) on the metabolites identified on HD4 as well as CLP platforms to assess whether doxorubicin treatment impacts tumor metabolism. The PCA revealed clear separation between the vehicle and doxorubicin treated tumors on both the HD4 (Figure 2C), and CLP (Figure 2D) platforms, suggesting the impact of doxorubicin on both global and lipid metabolism. The rather loose clustering observed among the biological replicates suggest metabolic variation between the samples.

To determine the metabolic pathways affected by the doxorubicin treatment, we conducted data analysis using our in-house developed tool, Autonomics, deploying linear model limma. We identified nominally significant (*p*-value ≤ 0.05) alteration after doxorubicin treatment in 127 metabolites measured on the HD4 platform. The doxorubicin treatment triggered changes in the metabolism of amino acids, carbohydrates, cofactors and vitamins as well as nucleotides and lipids. Among the lipids measured on CLP, we observed a nominally significant (*p*-value ≤ 0.05) decrease in 96 molecules after the treatment. These results indicate that doxorubicin treatment triggers metabolic rewiring, which could potentially contribute to cancer cell survival and acquired resistance. 

### 2.3. Doxorubicin Treatment Suppresses Glycolysis and Nucleotide Synthesis

Glucose and glutamine are recognized as two main cancer cell metabolic dependences [31], therefore we explored whether doxorubicin treatment impacted any of those pathways in the *in ovo* grafted tumors. 

We found that glucose level was elevated (Figure 3A) and the products of glycolysis such as fructose 1,6-diphosphate as well as lactate (Figure 3B,C) were decreased in doxorubicin-treated tumors. Doxorubicin treatment did not trigger alteration in the level of the glutamine (Figure 3D). 

The products of glucose and glutamine catabolism can serve as substrates for the biosynthesis of nucleotides in highly proliferative cells [31]. We found that almost all metabolites of purine (xanthine, inosine, adenine, and guanine) and pyrimidine (uracil and thymine) metabolism were decreased in doxorubicin-treated tumors (Table 1). Among all measured nucleotides, we observed an increase only in the levels of orotate (Figure 3E), deoxycytidine (Figure 3J), and 5-methylcytidine. Orotate, which depends on glutamine as a nitrogen source, together with 5-phosphoribosyl diphosphate (PRPP) (Figure 3F, from the pentose phosphate pathway, serve as a substrate for uridine metabolism [32]. The elevated level of orotate (Figure 3E), together with unchanged levels of PRPP (Figure 3F), decreased levels of uridine 5’-diphosphate (UDP) (Figure 3G), and uridine (Figure 3H) as well as cytidine diphosphate (Figure 3I), suggest suppression of uridine synthesis and metabolism. Taken together, these observations suggest that doxorubicin treatment suppressed glycolysis and interfered with nucleotide metabolism, resulting in decreased purine and pyrimidine synthesis. 

### 2.4. The Cholinic Phenotype and Fatty Acid Metabolism Are Suppressed by Doxorubicin Treatment

Accumulation of glycerophospholipids such as phosphatidylcholines and phosphatidylethanolamine is recognized as one of the hallmarks of dysregulated cancer cell metabolism and is defined as a cholinic phenotype, independent of the proliferative capacity of cancer cells [16]. A previous study reported alterations in choline metabolism triggered by doxorubicin treatment in breast cancer cell lines (MB-MDA-231 and MCF7) [33]. To assess whether similar responses could be observed *in ovo,* we investigated the impact of doxorubicin treatment on cholinic phenotype in our study. We found that the pathway of phosphatidylcholine and phosphatidylethanolamine synthesis, known as the Kennedy pathway [33], was suppressed under doxorubicin treatment *in ovo* (Figure 4). Metabolites contributing to phosphatidylcholine synthesis including choline (Figure 4A), and cytidine 5’-diphosphocholine (Figure 4B) as well as phosphatidylethanolamine synthesis including phosphoethanolamine (Figure 4D) and cytidine-5’-diphosphoethanolamine (Figure 4E) were lower in tumors treated with doxorubicin than in vehicle-treated tumors. Cytidine 5’-diphosphocholine and cytidine-5’-diphosphoethanolamine, together with 1,2-diacylglycerols, are used as substrates for phosphatidylcholine and phosphatidylethanolamine, respectively. The levels of 1,2-diacylglycerols as well as phosphatidylcholines and phosphatidylethanolamines were lower in doxorubicin-treated tumors (example Figure 4C,F). Among the molecules measured on the lipidomics platform, the majority of the altered metabolites (56 out of 95) were glycerophospholipids (phosphatidylcholines and phosphatidylethanolamine), which were all decreased (Table 2).

Fatty acids may be released from the glycerophospholipids to serve as signaling molecules or an energy source to promote oncogenic transformation and sustain energetic demands [16,23]. Following doxorubicin treatment, we observed a decrease in the levels of free fatty acids (e.g., Figure 4G) as well as acylcarnitines with different chain lengths (example Figure 4H), suggesting that fatty acid release from the glycerophospholipids is suppressed under the given treatment (Table 1). We also observed a decrease in the levels of glycerophosphorylcholine (Figure 4I) and glycerophosphoethanolamine (Figure 4J), which are breakdown products of phosphatidylcholine and phosphatidylethanolamine, respectively. These results suggest that cholinic phenotype as well as fatty acid metabolism is affected by doxorubicin treatment.

### 2.5. Pathways Supporting Tumor Antioxidative Capacity Are Activated in Response to Doxorubicin Treatment

A previous study suggested that tumor resistance to chemotherapeutics could be linked with the antioxidative capacity of the cells [34]. We further asked whether any of the metabolic dysregulations identified in doxorubicin−treated tumors could support their antioxidative capacity.

We found that ascorbate (Figure 5A), a potent antioxidant, among all measured metabolites, showed the greatest decrease in doxorubicin−treated tumors. The ascorbate metabolites including dehydroascorbate (Figure 5B) and oxalate (Figure 5C) were also altered. The decrease in ascorbate with simultaneous increase of its breaking product oxalate, suggest accelerated ascorbate metabolism, which might be linked with cellular response to oxidative stress driven by doxorubicin.

The antioxidative potential of the cell was recognized in the glutathione metabolism [35]. We found a decrease in reduced and oxidized glutathione (GSH (Figure 5D) and GSSG (Figure 5E)) levels as well as a decrease in the ratio of those metabolites (GSH/GSSG) (Figure 5F). Decrease in glutathione, with simultaneous increase in the level of cysteine−glutathione disulfide (Figure 5G), which is generated under the exposure to oxidative stress from GSH, suggest that antioxidative mechanisms, dependent on GSH, were activated.

The level of glutamate, a component in GSH synthesis, was lower in doxorubicin−treated tumors in comparison with DMSO (Figure 5H). Glutamate can be obtained directly from glutamine catabolism in the reaction catalyzed by glutaminase (GLS) or as a byproduct of orotate synthesis or branched chain amino acid (BCAA) catabolism [12]. The elevated level of orotate (Figure 3E) might suggest enhanced synthesis of the glutamate as a byproduct. We also observed increased levels of 3-methyl-2-oxovalerate (Figure 5I), 3-methyl-2-oxobutyate (Figure 5J) and 4-methyl-2-oxopentanoate (Figure 5K), which are products of the BCAA catabolic pathway obtained together with glutamate as a byproduct. The BCAA level remained unchanged under doxorubicin treatment, however, we observed a decrease in the level of dipeptides including leucylglycine, alanylleucine, leucylglycine, glycylleucine, and glycylvaline (Table 1), which can contribute to the BCAA pool. These data suggest that pathways contributing to the antioxidative capacity of the cell are upregulated under the doxorubicin treatment, potentially to protect the tumor from oxidative stress.

## 3. Discussion

This study was conducted to assess the feasibility of using tumor grafted *in ovo*, utilizing metabolic analysis as a strategy for future development of a robust platform for anticancer drug testing. We further aimed to provide insight into cancer survival mechanisms activated under doxorubicin treatment.

The *in ovo* model is a well−established system to study tumor biology, which has significantly contributed to our understanding of tumorigenesis [25,36] as well as anticancer drug testing [28]. This model has a significant advantage over 2D cell culture and organoids as it resembles the tumor microenvironment including tumor vascularization by chick vessels [24]. Currently, patient−derived xenografts (PDX) are tested *in ovo* as a more robust strategy, in context of cost and time efficiency, than mice [24], and future large-scale studies deploying *in ovo* model for PDX drug testing could be envisioned. Thus, development of a broader portfolio for *in ovo* readout of tumor responses to tested conditions is of great importance.

Metabolomics, previously recognized as a crucial player in the advancement of personalized medicine [37], could be of significant relevance for the monitoring of drug responses of tumors grafted *in ovo* and could further aid in the optimization of treatment strategies. In this study, we showed the feasibility of deploying metabolomics to monitor responses of *in ovo* grafted tumors to drug treatment, which to the best of our knowledge, is the first study of such kind. We showed evident separation, based on the metabolic profiles, measured on global untargeted metabolic and lipidomic platforms, between doxorubicin and DMSO−treated tumors, which support the feasibility of such an approach. The rather loose distribution within the cluster, observed on the PCA plot, suggest metabolic variability between the samples, which might be driven by metabolic differences between the eggs. The metabolic variation between the samples from the same group further impacts the significance of the observed differences. This could be overcome by scaling up the number of samples per group.

The metabolic alterations, identified as one of the hallmarks of cancer [6], were later linked with oncogenic signaling and acquired resistance to treatment [38]. Although, doxorubicin-treated tumors were found to be significantly smaller in size than the control or DMSO-treated tumors in our study, the growth was rather suppressed and not completely inhibited. Thus, the cell population, which under doxorubicin treatment formed tumor mass, potentially acquired resistance. Metabolomics profiling has been proven as a successful strategy for the identification of cancer survival mechanism and treatment optimization [39,40]. Thus, the metabolic alterations, observed under doxorubicin treatment in *in ovo* grafted tumors could also point toward cancer survival mechanisms.

The alterations in glycolysis, nucleotide synthesis, and choline metabolism could be associated with growth suppression, which we found in doxorubicin-treated tumors. The alteration in cholinic phenotype, observed in our study, is in good agreement with previous reports conducted in 2D cell culture focusing on choline metabolism under doxorubicin treatment as a predictive measure of treatment responses [33].

Our observation of elevated glucose level in the tumor after the doxorubicin treatment might suggest the inhibition of glycolytic processes, which may be linked with the inhibition of cell proliferation and biomass demand or interference of doxorubicin with glucose metabolism. The systemic insulin resistance, mimicking type 2 diabetes phenotype, was associated with hyperglycemia and insulin resistance triggered in skeletal muscle by doxorubicin treatment [31,41]. This observation would require further investigation, focusing on combining doxorubicin and blood glucose normalizing components such as metformin or SGL2 inhibitors to assess whether normalized blood glucose impacts treatment responses. In fact, a clinical trial combining doxorubicin and metformin was initiated in breast cancer patients (NCT02506777/Active−Recruiting Aug 2020) to further elucidate whether such a combination would improve patient outcome.

The metabolic alterations in ascorbate and glutathione metabolism could reflect on the antioxidative capacity of the tumors, enabling the oxidative stress to be overcome, triggered by doxorubicin, and thus the development of tumor masses. A previous study reported that exposure of cancer cells to high ascorbate concentrations effectively and selectively eliminated KRAS and BRAF mutant cancer cells due to the uptake of an oxidized form of ascorbate (dehydroascorbate (DHA)), which in the cell is reduced to ascorbate and causes oxidative stress [42]. In our study, levels of ascorbate as well as DHA were lower and the product of their metabolism, oxalate, was elevated in doxorubicin-treated tumors, suggesting activation of ascorbate metabolism in tumor to manage oxidative stress. Therefore, there is a possibility of improving the outcome of patients carrying KRAS and BRAF mutations by supplementing chemotherapy with ascorbate to maximize the oxidative stress in tumor cells. Nevertheless, KRAS and BRAF mutations are rare in TNBC and supplementation with ascorbate might not benefit these patients [43].

The accelerated glutathione metabolism, supported by glutamate, potentially delivered from multiple pathways including glutaminolysis, glycolysis, and BCAA catabolism, could be considered as crucial factors to cancer survival under the treatment and tumor formation. Previous studies have suggested a significant role of glutathione metabolism in cancer resistance to chemotherapeutics [11,34]. Thus, strategies aiming to suppress the antioxidative potential of the tumor could improve the treatment. The intervention in glutamate metabolism by glutaminase inhibitors was shown to impact the glutathione level [15,39] and thus, reduce the antioxidative tumor capacity. Indeed, strategies investigating combination of glutaminase inhibitor, CB−838, with chemotherapeutics such as paclitaxel and azacitidine are already the subjects of clinical trials (NCT03057600 and NCT03047993). However, we showed that under doxorubicin treatment, glutamate might potentially be obtained from BCAA catabolism and glycolysis to support the glutathione synthesis in MB−MDA−231 grafted *in ovo*. Therefore, strategies directly targeting glutathione metabolism in combination with doxorubicin might sensitize tumors to doxorubicin treatment. Buthionine sulfoximine (BSO), a potent inhibitor of gamma−glutamylcysteine synthetase, catalyzing glutathione synthesis [44], was shown to synergistically enhance melphalan activity against preclinical models of multiple myeloma in the xenograft model [45]. Nevertheless, oxidative stress was established as a main factor of cardiotoxicity induced by doxorubicin treatment [46], and thus strategies targeting the antioxidative capacity of cancer in combination with chemotherapeutic agents would require careful monitoring.

In conclusion, we have shown here for the first time the feasibility of an implementation of metabolomics for anticancer drug testing in an *in ovo* tumor model. This is crucial for the future establishment of a comprehensive and robust platform, which may serve for the testing of new components as well as for the optimization of treatment strategies in patient derived xenografts *in ovo*, to be used in personalized medicine. Moreover, we have revealed that metabolic processes altered in tumor under doxorubicin treatment, suggesting a significant role of antioxidative pathways in cancer cell survival under the treatment. Combination of doxorubicin with metabolic agents targeting antioxidative capacity of the cell might sensitize tumors to doxorubicin and would require further investigation.

## 4. Materials and Methods

### 4.1. In Ovo Experiments

All activities related to the *in ovo* experiments including the preparation of the final formulation of compounds used in this study, the grafting of MB−MDA−231 cell line on CAM, the incubation of the eggs, the administration of test compounds, the toxicity test, and the tumor growth analysis were carried out by the INOVOTION lab team (INOVOTION, France). All activities related to the *in ovo* experiments at INOVOTION were conducted as previously described [26]. According to French legislation, no ethical approval is needed for scientific experiments using oviparous embryos (decree no. 2013−118, February 1, 2013; art. R−214−88).

#### 4.1.1. Chick Embryo Tumor Grafting and Treatment

Fertilized White Leghorn eggs were incubated at 37.5 °C with 50% relative humidity over nine days. At day 9 of embryonal development (E9), a small hole was drilled through the eggshell into the air sac, and 1 cm^2^ window was cut in the eggshell above the CAM. At least twenty-five eggs were used per condition to account for the survival rate after the first 9 days of development.

The MDA−MB−231 cell line was cultivated in DMEM medium supplemented with 10% fetal bovine serum (FBS) and 1% penicillin/streptomycin. The cells were detached by trypsinization, washed with DMEM complete medium, and suspended in graft medium (DMEM without FBS). A sample of 50 µL of 20 × 10^6^/mL MDA−MB−231 cell suspension (corresponding to 1 × 10^6^ cells) was used for inoculation onto the CAM of each egg. Eggs were randomized into three groups including untreated control, DMSO-treated tumors, and doxorubicin-treated tumors. On day 10 of the embryo development (E10), tumors began to be detectable. The treatment started at E11 by dropping 100 μL of doxorubicin (50 μM) or vehicle (1% of DMSO in phosphate buffer saline (PBS)) onto the tumor. The treatment was repeated at E13, E15, and E17.

#### 4.1.2. Tumor Growth Analysis

The tumor sample collection and growth analysis were conducted at E18. A total of three replicates per group were used for the growth analysis. The upper portion of the CAM containing the tumor was removed, washed in PBS, and incubated with paraformaldehyde for 48 h. The tumor was washed in PBS, and carefully cut off from normal CAM tissue. The obtained tumors were imaged and weighed.

#### 4.1.3. Fresh Tumor Collection for Metabolic Analysis

The fresh tumor collection was conducted at E18. A total of five replicates per group was used for the metabolic analysis. The upper portion of the CAM containing tumor was removed, and the tumor was washed briefly in phosphate-buffered saline (PBS). The tumor was placed in a pre-cooled Eppendorf tube and directly frozen in liquid nitrogen. The tumor samples were sent on dry ice for the metabolic measurements.

### 4.2. Metabolic Profiling

The measurements were conducted on tumor samples using two different Metabolon platforms including the HD4 platform and the complex lipid platform (CLP). The sample preparation was conducted using an automated MicroLab STAR system (Hamilton). The tumor samples were extracted using the optimized Bligh−Dyer extraction method deploying methanol/water/dichloromethane as an extraction solvent [47]. The extraction was conducted in the presence of internal standards. For each platform, a separate homogenized aliquot was submitted. A portion of ~15 µL from each sample extract was obtained and pooled to prepare the Client Matrix (CM). The CM samples were processed in triplicate along with other samples. The instrument variability was determined by calculating the median relative standard deviation (RSD) for the internal standards that were added to each sample prior to injection into the mass spectrometers. The process variability was determined by calculating the median RSD for all endogenous metabolites (i.e., non−instrument standards) present in 100% of the CM samples.

#### 4.2.1. Non-Targeted Metabolic Profiling (HD4 Platform)

The samples’ extracts were divided into equal fractions, dedicated for the measurements at different analytical platforms, and were evaporated and stored under nitrogen stream overnight (TurboVap (Zymark)) [29]. Each sample fraction was reconstituted in four different solvents optimized for the given measurement method as previously described [29,39]. (1) The acidic positive ion conditions for the hydrophilic compounds were as follows: the extract was gradient eluted from a C18 column (Waters UPLC BEH C18−2.1 × 100 mm, 1.7 µm) using water and methanol containing 0.05% perfluoropentanoic acid (PFPA) and 0.1% formic acid (FA) and analyzed using mass spectrometry (MS)/MS methods with positive ion mode electrospray ionization (ESI). (2) The acidic positive ion conditions for the hydrophobic compounds were as follows: the extract was gradient eluted from a C18 column (Waters UPLC BEH C18−2.1 × 100 mm, 1.7 µm) using methanol, acetonitrile, water, 0.05% PFPA, and 0.01% FA, and analyzed using MS/MS methods with positive ion mode ESI. (3) The basic negative ion conditions were: the extract was gradient eluted from the column C18 (Waters UPLC BEH C18−2.1 × 100 mm, 1.7 µm) using methanol and water containing 6.5 mM ammonium bicarbonate at pH 8 and analyzed using MS/MS methods with negative ion mode ESI. (4) The extract was gradient eluted from the HILIC column (Waters UPLC BEH Amide 2.1 × 150 mm, 1.7 µm) using water and acetonitrile with 10 mM ammonium formate, pH 10.8, and analyzed using negative ion mode ESI.

All measurements were conducted by deploying Waters ACQUITY ultra liquid chromatography (UPLC) and Thermo Scientific Q Exactive high resolution/accurate mass spectrometer interfaced with a heated electrospray ionization (HESI−II) source and Orbitrap mass analyzer operated at 35,000 mass resolution [29]. The instrument and process variability related to HD4 platform was 4% and 9%, respectively.

#### 4.2.2. Lipidomic Profiling (CLP Platform)

The portion of the sample extract dedicated to lipidomic profiling was dried under nitrogen flow and reconstituted in ammonium acetate dichloromethane:methanol. The extracts’ infusion was performed on a Shimadzu LC with nano PEEK tubing as previously described [30]. The samples were analyzed in both positive and negative mode electrospray using QTRAP 5500 system with the SelexION device (Sciex). The molecules were detected in multiple reaction monitoring (MRM) mode with a total of more than 1100 MRMs. Individual lipid species were quantified by the ratio of the signal intensity of each target compound to that of its assigned internal standard, followed by the multiplication of the concentration of internal standard added to the sample. Lipid class concentrations were calculated from the sum of all molecular species within a class, and fatty acid compositions were determined by calculating the proportion of each class comprised of individual fatty acids, as previously described [30,48]. The process variability found on CLP was 8%.

### 4.3. Viability Assay

The MB−MDA−231 was obtained from the American Type Culture Collection (ATCC, Manassas, VA, USA) and cultivated in Roswell Park Memorial Institute medium (RPMI-1640) media supplemented with 10% FBS and 1% penicillin-streptomycin solution, as recommended by the cell line provider. The cells were seeded in a 96-well plate at a density of 17,000 cells/well and incubated at 37 °C in a humidified atmosphere of 5% CO_2_. Following 24 h incubation, the cells were treated with vehicle (DMSO) or two different doxorubicin concentrations (1 µM and 5 µM); the untreated cells were used as a control. The viability of the cells was assessed after the treatment at the following time points 24 h, 48 h, and 72 h using the 3-(4,5-dimethylthiazol-2-yl)-2,5-diphenyltetrazolium (MTT) assay [8]. The MTT solution (1.2 mM) was added to each well and incubated at 37 °C for 2 h to enable the formation of purple formazan crystals. The supernatant was aspirated and the plate was stored at −80 °C until measured. On the day of the measurement, the plates were removed from the freezer and kept at room temperature (RT) for 45 min. To each well, 200 µL DMSO was added and the samples were incubated at RT under constant shaking for 1 h. The absorbance was measured at 570 nm using a CLARIOstar^®^ High Performance Monochromator Multimode Microplate Reader (BMG LABTECH, Ortenberg, Germany).

### 4.4. Statistical Data Analysis

We used run-day median-scaled metabolite values as provided by Metabolon. Data analysis was performed using R studio (R version 3.5.3) and our in-house developed tool Autonomics (https://github.com/bhagwataditya/autonomics). Using an Autonomics pipeline, data were log-transformed and fitted to the “~ 0 + subgroup" model using the limma package. We extracted the metabolites with a significant differential effect (*p* < 0.05) for the contrast “Vehicle treated-Doxorubicin treated”, and box plots were generated.

Principal component analysis (PCA) was performed with SIMCA version 15 (Umetrics, Umea, Sweden).

## Figures and Tables

**Figure 1 metabolites-10-00268-f001:**
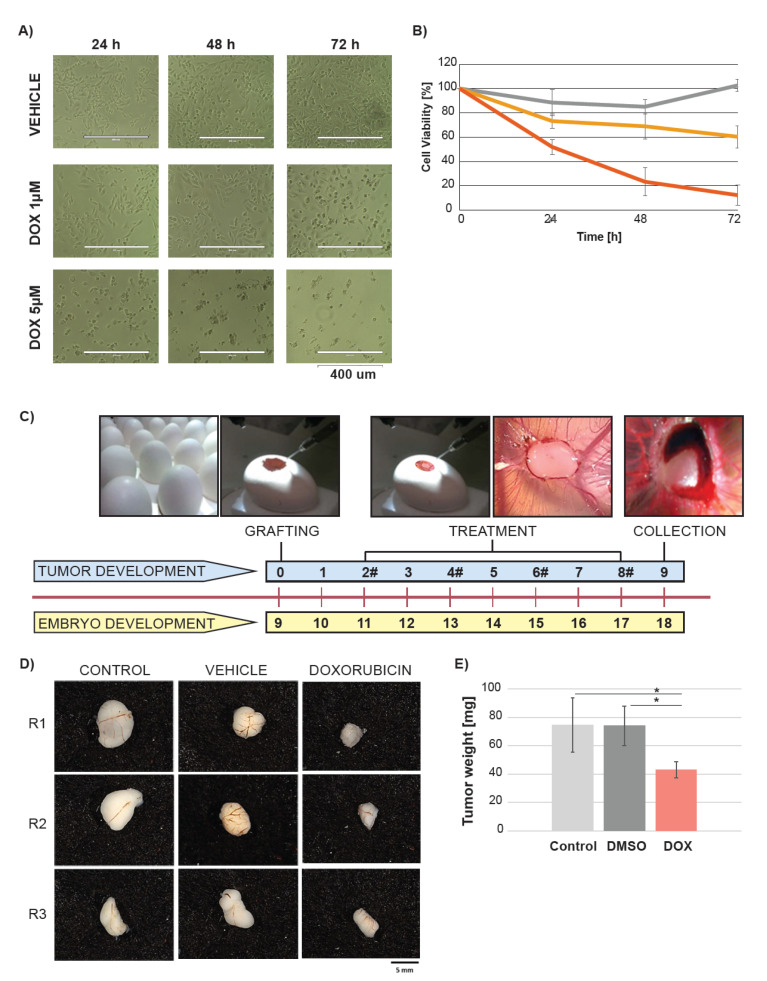
Doxorubicin decreases cell viability *in ovo* and in vitro. (**A**) Representative appearance of the MB-MDA-231 cultured in vitro over time of 72 h after treatment with vehicle (DMSO) and two doxorubicin (DOX) concentrations. (**B**) Cell viability after the DMSO or DOX treatment depicted by the (3-(4,5-dimethylthiazol-2-yl)-2,5-diphenyltetrazolium bromide) MTT assay. Dark grey indicates vehicle, yellow and red indicate 1 µM and 5 µM concentration of DOX respectively. (**C**) Study design. The chick embryo grew over a period of nine days. At day 9 of embryo growth, the cancer cells from the MB-MDA-231 cell line were grafted on the CAM of the egg. The treatment with DMSO, and DOX started at day 2 after cancer cell grafting. A volume of 100 µL of 25 µM DOX was added to achieve 0.024 mg/km of DOX per egg. The untreated cells were used as a control. The treatment was conducted every two days and the treatment time points are depicted by #. The tumors were collected at day 9 after the cancer cell grafting. (**D**) Representative gross appearance of tumors excised from the chick embryo (each group presented in triplicates). (**E**) Impact of DOX on tumor weight. Light grey indicates control, dark grey indicates vehicle, and red indicates DOX treatment. Significant differences were depicted by *.

**Figure 2 metabolites-10-00268-f002:**
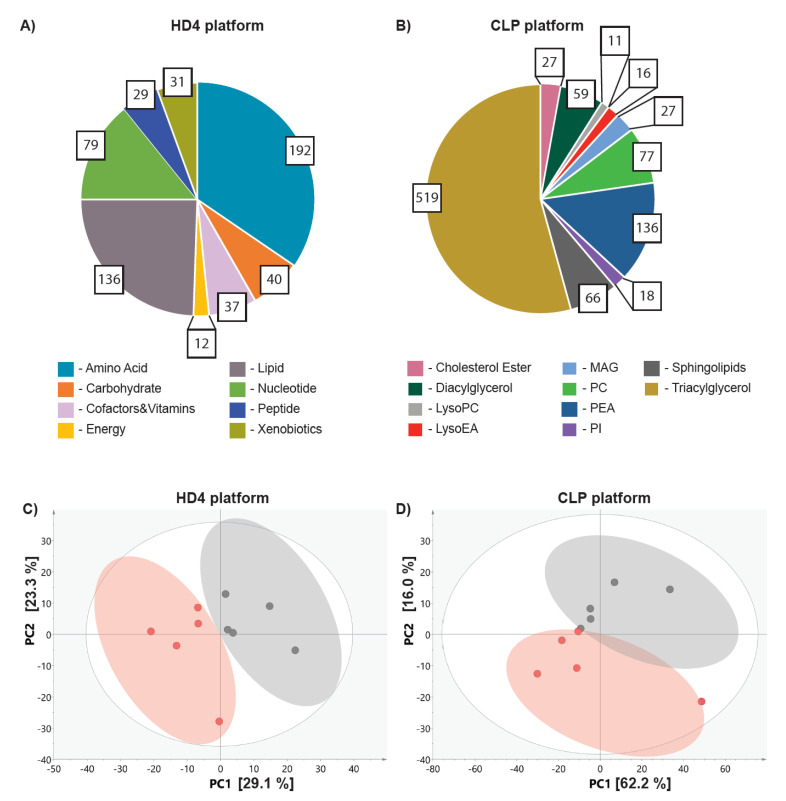
Doxorubicin triggers changes in tumor metabolism *in ovo*. Pie chart reflective of the number of metabolites measured on HD4 platform (**A**) and CLP (**B**), representing the numerical proportion of each metabolic class. The colors of the pie fractions represent different metabolic classes measured on the HD4 or CLP platform. The PCA analysis of metabolites measured on HD4 (**C**) and CLP (**D**) reveal the separation between the vehicle (depicted in grey) treated and doxorubicin (depicted in red).

**Figure 3 metabolites-10-00268-f003:**
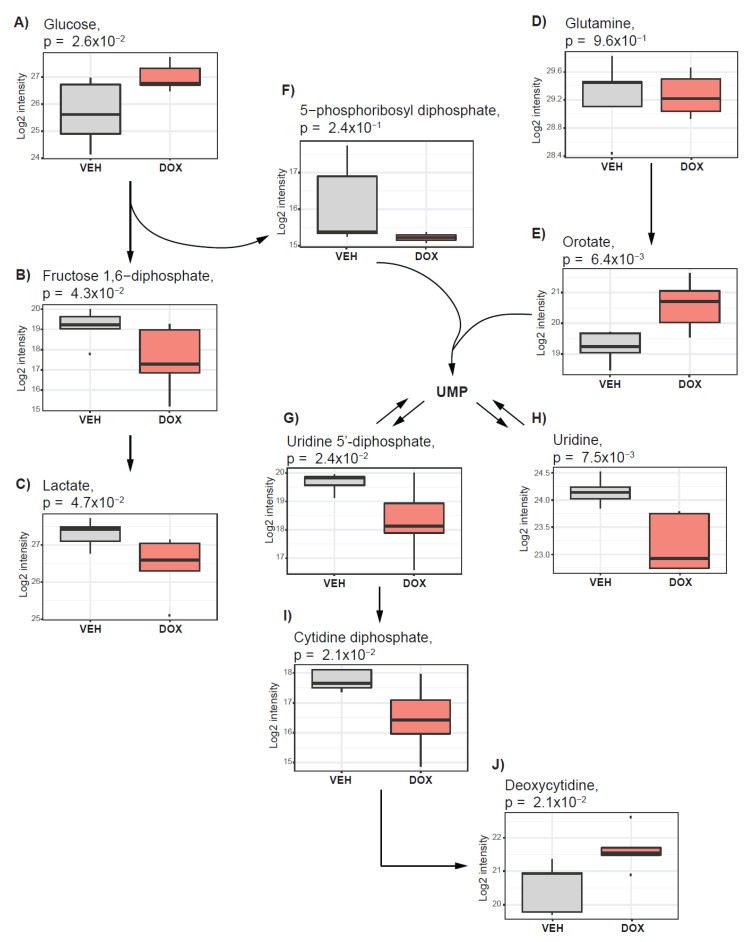
Doxorubicin treatment suppresses glycolysis and nucleotide synthesis. The boxplots represent alterations in glycolysis and nucleotide synthesis after treatment of *in ovo* tumors with doxorubicin. Alterations in (**A**) Glucose, (**B**) Fructose 1,6-diphosphate, (**C**) lactate, (**E**) Orotate, (**G**) Uridine 5’-diphosphate, (**H**) Uridine, (**I**) Cytidine diphosphate, and (**J**) Deoxycytidine are nominally significant. (**D**) Glutamine and (**F**) 5-phosphoribosyl-diphosphate are not impacted by doxorubicin treatment. Vehicle-treated tumors (VEH) are depicted in grey, and those treated with doxorubicin (DOX) are indicated in red.

**Figure 4 metabolites-10-00268-f004:**
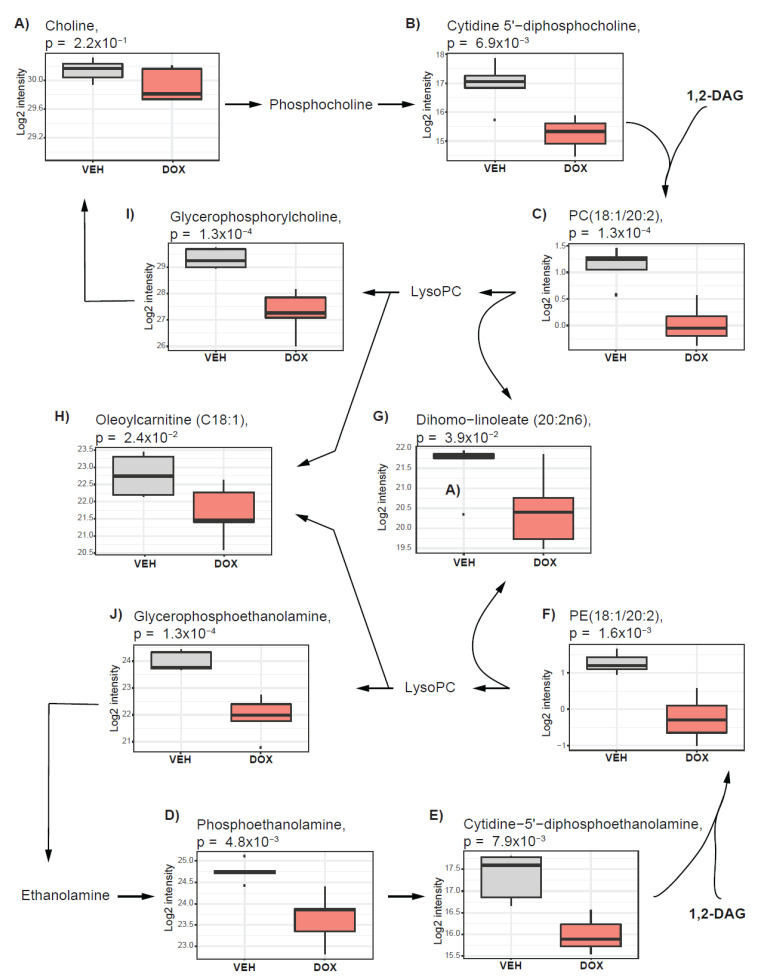
Cholinic phenotype is impacted by doxorubicin treatment *in ovo***.** The boxplots present the alteration for glycerophospholipid metabolism triggered after the treatment of *in ovo* tumors with doxorubicin. (**A**) Choline level is not affected by doxorubicin treatment. (**B**) Cytidine 5’−diphosphocholine, (**C**) phosphatidylcholines (e.g., PC(18:1/20:2)), (**D**) Phosphoethanolamine, (**E**) Cytidine−5’−diphosphoethanolamine, (**F**) Phosphoethanolamines (e.g., PE(18:1/20:2)), (**G**) Dihomo−linoleate (20:2n6), (**H**) Oleoylcarnitine (C18:1), (**I**) Glycerophosphorylcholine, and (**J**) Glycerophosphoethanolamine show nominally significant alterations after the treatment. Vehicle-treated tumors (VEH) are depicted in grey, and those treated with doxorubicin (DOX) are indicated in red.

**Figure 5 metabolites-10-00268-f005:**
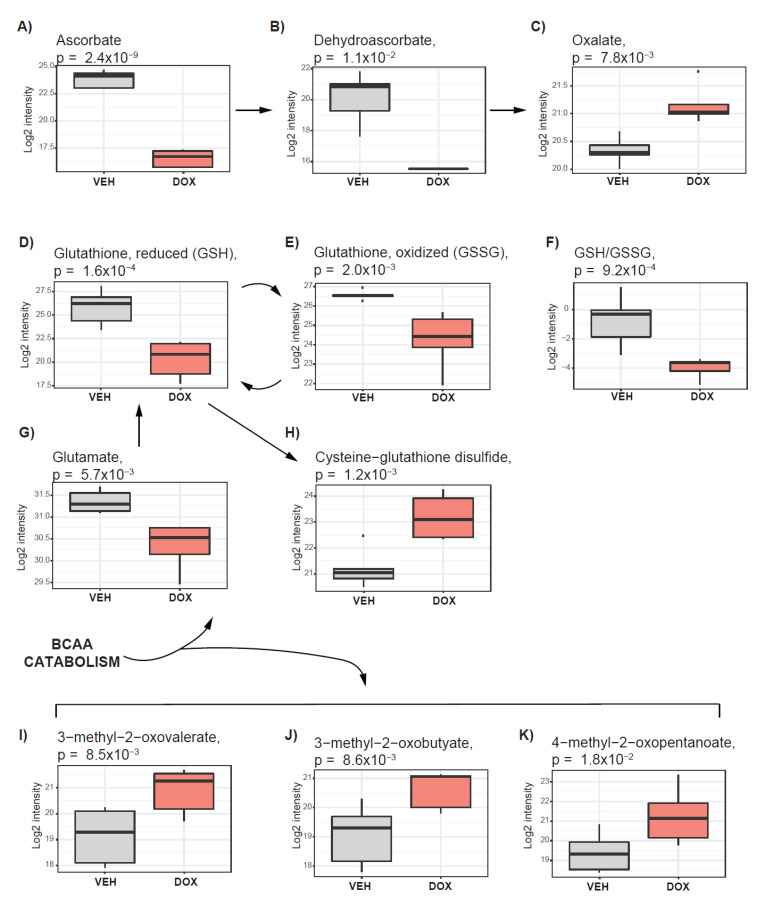
Activation of pathways supporting tumor antioxidative capacity in response to doxorubicin treatment. The boxplots represent alterations in ascorbate metabolism (**A**–**C**); Glutathione metabolism (**D**,**E**,**G**,**H**); metabolic ratio between reduced and oxidized glutathione (GSH/GSSG) (**F**); Branch chain amino acid catabolism (**I**–**K**). Vehicle−treated cells are depicted in grey, and those treated with doxorubicin are indicated in red.

**Table 1 metabolites-10-00268-t001:** Metabolic alterations driven by doxorubicin treatment *in ovo* determined on a global untargeted metabolomics platform.

Metabolite	Pathway	Sub-Pathway Metabolism	Beta	*p*-Value
*N*-acetylaspartate (NAA)	Amino Acid	Alanine and Aspartate	−2.50	1.08 × 10^−3^
*N*-acetylalanine			−1.25	1.49 × 10^−3^
*N*-acetylglutamate		Glutamate	−3.10	1.67 × 10^−4^
*N*-acetyl-aspartyl-glutamate			−2.25	3.73 × 10^−3^
Beta-citrylglutamate			−1.65	4.68 × 10^−3^
Glutamate			−1.02	5.69 × 10^−3^
Glutathione, reduced (GSH)		Glutathione	−5.53	1.63 × 10^−4^
Glutathione, oxidized (GSSG)			−2.31	2.05 × 10^−3^
*S*-(1,2-dicarboxyethyl)glutathione			−2.01	2.76 × 10^−3^
*S*-methylglutathione			−1.45	9.82 × 10^−3^
Cysteine-glutathione disulfide			2.00	1.16 × 10^−3^
*N*-acetylserine		Glycine, Serine, and Threonine	−0.74	1.68 × 10^−2^
Glycine			−0.66	3.47 × 10^−2^
*N*-acetylglycine			1.02	1.15 × 10^−2^
1-methylhistamine		Histidine	−1.39	2.95 × 10^−3^
N-acetylhistamine			−1.26	3.00 × 10^−2^
Imidazole lactate			0.90	4.47 × 10^−2^
3-methyl-2-oxobutyrate		Leucine, Isoleucine, and Valine	1.57	8.59 × 10^−3^
3-methyl-2-oxovalerate			1.75	8.47 × 10^−3^
4-methyl-2-oxopentanoate			1.87	1.80 × 10^−2^
N2-acetyllysine		Lysine	−0.98	3.23 × 10^−3^
Fructosyllysine			0.80	4.47 × 10^−2^
*N*-acetylmethionine		Methionine, Cysteine, SAM, and Taurine	−1.71	5.53 × 10^−4^
*S*-adenosylmethionine (SAM)			−1.20	6.85 × 10^−3^
*N*-acetylmethionine sulfoxide			−0.71	3.39 × 10^−2^
Hypotaurine			−0.85	3.77 × 10^−2^
*N*(’1)-acetylspermidine		Polyamine	−1.96	2.82 × 10^−3^
Spermidine			−0.71	1.55 × 10^−2^
*N*-acetylputrescine			−0.91	3.19 × 10^−2^
N1,N12-diacetylspermine			2.67	1.16 × 10^−2^
Serotonin		Tryptophan	−1.29	1.14 × 10^−2^
*N*-formylanthranilic acid			1.34	6.32 × 10^−3^
*O*-methyltyrosine		Tyrosine	−1.24	9.36 × 10^−3^
1-carboxyethyltyrosine			−1.13	1.54 × 10^−2^
*N*-formylphenylalanine			0.89	2.06 × 10^−2^
*N*-monomethylarginine		Urea cycle	−1.54	3.10 × 10^−2^
*N*-acetylglucosaminylasparagine	Carbohydrate	Aminosugar	−0.95	3.57 × 10^−3^
*N*-acetylglucosamine 6-P			−1.03	2.32 × 10^−2^
*N*-acetylneuraminate			−0.73	3.75 × 10^−2^
Glucose 1,6-diphosphate		Glycolysis, Gluconeogenesis, and Pyruvate	−1.62	4.27 × 10^−2^
Lactate			−0.86	4.71 × 10^−2^
Glucose			1.33	2.56 × 10^−2^
UDP-glucose		Nucleotide Sugar	−1.50	2.35 × 10^−3^
UDP-glucuronate			−0.79	8.12 × 10^−3^
UDP-galactose			−2.28	1.41 × 10^−2^
UDP-*N*-acetylglucosamine			−0.97	2.46 × 10^−2^
Ascorbate (Vitamin C)	Cof & Vit.	Ascorbate and Aldarate	−7.29	2.44 × 10^−9^
Dehydroascorbate			−4.57	1.10 × 10^−2^
Oxalate (ethanedioate)			0.83	7.79 × 10^−3^
Nicotinamide		Nicotinate and Nicotinamide	−1.42	1.76 × 10^−3^
Nicotinamide adenine dinucleotide reduced (NADH)			−1.43	9.38 × 10^−3^
Nicotinamide adenine dinucleotide (NAD+)			−1.00	9.61 × 10^−3^
Nicotinate			−0.80	9.71 × 10^−3^
Adenosine 5’-diphosphoribose			−1.10	2.92 × 10^−2^
Nicotinamide ribonucleotide			1.22	1.71 × 10^−2^
Pantothenate		Pantothenate and CoA	−1.14	7.54 × 10^−4^
Flavin adenine dinucleotide (FAD)		Riboflavin	−0.72	3.53 × 10^−2^
Thiamin (Vitamin B1)		Thiamine	−1.35	2.91 × 10^−3^
Alpha-tocopherol		Tocopherol	−2.30	2.76 × 10^−2^
Pyridoxal phosphate		Vitamin B6	−1.18	2.50 × 10^−3^
Pyridoxamine phosphate			−1.83	3.79 × 10^−3^
Pyridoxamine			−0.99	3.07 × 10^−2^
Phosphate	Energy	Oxidative Phosphorylation	−0.61	2.24 × 10^−2^
Succinate		TCA Cycle	−1.03	1.01 × 10^−2^
Malate			−0.86	1.63 × 10^−2^
Deoxycarnitine	Lipid	Carnitine	−1.09	2.94 × 10^−3^
Docosahexaenoyl ethanolamide		Endocannabinoid	−0.74	1.51 × 10^−2^
Arachidoylcarnitine (C20)		Fatty Acid (Acyl Carnitine)	−1.19	3.11 × 10^−2^
Palmitoleoylcarnitine (C16:1)			−1.24	1.80 × 10^−2^
Oleoylcarnitine (C18:1)			−1.09	2.40 × 10^−2^
Arachidonoylcarnitine (C20:4)			−1.89	5.38 × 10^−3^
Linoleoylcarnitine (C18:2)			−1.91	8.02 × 10^−3^
Acetylcarnitine (C2)			−0.72	4.67 × 10^−2^
Butyrylcarnitine (C4)			−0.83	2.34 × 10^−2^
Glycerophosphoglycerol		Glycerolipid	−0.77	1.29 × 10^−2^
Glycerol			−0.68	2.10 × 10^−2^
Myo-inositol		Inositol	−1.35	2.29 × 10^−3^
Docosapentaenoate (22:5n3)		Long Chain Polyunsaturated Fatty Acid (n3 and n6)	−1.70	8.61 × 10^−3^
Tetradecadienoate (14:2)			−0.61	2.75 × 10^−2^
Dihomo-linoleate (20:2n6)			−1.10	3.85 × 10^−2^
Docosapentaenoate (22:5n6)			−1.42	4.75 × 10^−2^
Caproate (6:0)		Medium Chain Fatty Acid	0.98	7.18 × 10^−3^
3-hydroxy-3-methylglutarate		Mevalonate	−1.15	4.22 × 10^−3^
1-palmitoyl-2-oleoyl-GPG (16:0/18:1)		Phosphatidylglycerol (PG)	−1.16	3.95 × 10^−3^
1-stearoyl-2-oleoyl-GPS (18:0/18:1)		Phosphatidylserine (PS)	−0.84	9.50 × 10^−3^
Glycerophosphoethanolamine		Phospholipid	−2.05	1.30 × 10^−4^
Glycerophosphorylcholine (GPC)			−2.05	2.30 × 10^−4^
Phosphoethanolamine			−1.09	4.80 × 10^−3^
Cytidine 5’-diphosphocholine			−1.72	6.90 × 10^−3^
Cytidine-5’-diphosphoethanolamine			−1.34	7.86 × 10^−3^
Glycerophosphoinositol			−0.66	2.83 × 10^−2^
Inosine	Nucleotide	Purine ((Hypo)Xanthine/Inosine)	−1.30	1.46 × 10^−3^
Hypoxanthine			−0.82	1.22 × 10^−2^
Allantoic acid			−0.86	3.68 × 10^−2^
2’-deoxyinosine			−0.73	4.88 × 10^−2^
Adenylosuccinate		Purine (Adenine)	−1.66	3.99 × 10^−3^
Adenosine			−1.16	4.36 × 10^−3^
2’-deoxyadenosine			−1.07	4.83 × 10^−2^
Guanosine		Purine (Guanine)	−1.40	1.30 × 10^−3^
Guanine			−1.87	1.55 × 10^−3^
Guanosine 5’-diphosphate (GDP)			−1.56	2.36 × 10^−2^
2’-deoxyguanosine			−0.72	4.79 × 10^−2^
Cytidine diphosphate		Pyrimidine (Cytidine)	−1.29	2.30 × 10^−2^
Cytidine 5’-monophosphate			−0.66	2.41 × 10^−2^
2’-deoxycytidine			1.10	2.09 × 10^−2^
5-methylcytidine			1.53	1.38 × 10^−2^
Orotate		Pyrimidine (Orotate)	1.36	6.40 × 10^−3^
5,6-dihydrothymine		Pyrimidine (Thymine)	−0.76	2.21 × 10^−2^
Uridine 2’-monophosphate		Pyrimidine (Uracil)	−1.45	2.77 × 10^−3^
Uridine			−0.97	7.47 × 10^−3^
Uracil			−1.00	8.67 × 10^−3^
Uridine 5’-diphosphate (UDP)			−1.36	2.42 × 10^−2^
5-methyluridine (ribothymidine)			−0.83	4.42 × 10^−2^
Phenylacetylglycine	Peptide	Acetylated Peptides	−1.95	2.44 × 10^−4^
Glycylvaline		Dipeptide	−1.46	9.67 × 10^−4^
Glycylleucine			−1.45	1.39 × 10^−3^
Leucylglycine			−1.37	5.91 × 10^−3^
Phenylalanylglycine			−0.90	2.21 × 10^−2^
Alanylleucine			−1.13	2.64 × 10^−2^
Gamma-glutamylglutamate		Gamma-glutamyl Amino Acid	−1.48	2.01 × 10^−2^
Gamma-glutamylserine			0.82	1.86 × 10^−2^
Gamma-glutamylhistidine			0.89	3.67 × 10^−2^
Gamma-glutamyl-alpha-lysine			0.99	2.54 × 10^−2^
Gamma-glutamylthreonine			1.06	2.03 × 10^−2^
Gamma-glutamylalanine			1.17	2.73 × 10^−2^
Catechol sulfate	Xenobiotics	Benzoate	1.15	2.76 × 10^−2^
Ethyl glucuronide		Chemical	−1.84	1.02 × 10^−2^

**Table 2 metabolites-10-00268-t002:** Metabolic alterations driven by doxorubicin treatment *in ovo* determined on the lipidomics platform.

Metabolite	Pathway	Beta	*p*-Value
DAG(16:0/22:5)	Diacylglycerol Ester	−1.21	2.10 × 10^−2^
DAG(16:0/22:6)		−1.30	1.69 × 10^−2^
DAG(16:1/22:6)		−1.08	3.34 × 10^−2^
DAG(18:2/22:5)		−1.04	3.54 × 10^−2^
DAG(18:2/22:6)		−1.11	3.51 × 10^−2^
LPE(22:5)	Lysophosphatidylethanolamine Ester	−0.85	2.78 × 10^−2^
MAG(18:1)	Monoacylglycerol Ester	−0.87	1.01 × 10^−2^
PC(14:0/18:1)	Phosphatidylcholine Ester	−0.85	3.31 × 10^−2^
PC(16:0/14:0)		−1.48	2.27 × 10^−3^
PC(16:0/16:1)		−1.08	1.84 × 10^−2^
PC(16:0/18:0)		−0.79	3.42 × 10^−2^
PC(16:0/20:1)		−1.00	1.10 × 10^−2^
PC(16:0/20:2)		−1.07	7.59 × 10^−3^
PC(18:0/14:0)		−1.07	1.15 × 10^−2^
PC(18:0/16:1)		−1.09	3.49 × 10^−2^
PC(18:0/18:1)		−1.01	2.50 × 10^−2^
PC(18:1/16:1)		−1.04	1.12 × 10^−2^
PC(18:1/18:1)		−1.31	6.00 × 10^−3^
PC(18:1/18:2)		−0.76	3.81 × 10^−2^
PC(18:1/20:2)		−1.10	4.77 × 10^−3^
PC(18:1/20:3)		−0.83	3.02 × 10^−2^
PC(18:1/22:4)		−0.88	2.23 × 10^−2^
PC(18:1/22:6)		−0.88	1.69 × 10^−2^
PC(18:2/16:1)		−0.67	4.95 × 10^−2^
PC(20:0/18:1)		−0.84	3.37 × 10^−2^
PE(16:0/16:0)	Phosphatidylethanolamine Ester	−0.76	3.29 × 10^−2^
PE(16:0/18:1)		−0.73	4.69 × 10^−2^
PE(16:0/20:1)		−0.81	1.38 × 10^−2^
PE(16:0/20:2)		−1.23	1.76 × 10^−3^
PE(16:0/22:4)		−0.82	3.72 × 10^−2^
PE(16:0/22:6)		−0.89	3.35 × 10^−2^
PE(18:0/16:0)		−0.86	4.24 × 10^−2^
PE(18:0/18:1)		−1.19	7.65 × 10^−3^
PE(18:0/20:1)		−1.26	7.23 × 10^−3^
PE(18:0/20:2)		−1.56	6.21 × 10^−3^
PE(18:0/22:4)		−0.96	2.53 × 10^−2^
PE(18:0/22:5)		−1.30	7.27 × 10^−3^
PE(18:0/22:6)		−1.31	7.55 × 10^−3^
PE(18:1/16:1)		−0.98	1.30 × 10^−2^
PE(18:1/18:1)		−1.46	3.23 × 10^−3^
PE(18:1/20:1)		−1.58	9.29 × 10^−4^
PE(18:1/20:2)		−1.52	1.59 × 10^−3^
PE(18:1/20:3)		−1.23	1.38 × 10^−2^
PE(18:1/20:4)		−0.80	3.89 × 10^−2^
PE(18:1/22:4)		−1.41	3.15 × 10^−3^
PE(18:1/22:5)		−1.48	1.83 × 10^−3^
PE(18:1/22:6)		−1.46	3.45 × 10^−3^
PE(18:2/22:4)		−0.54	4.71 × 10^−2^
PE(O-18:0/16:0)		−1.19	9.90 × 10^−3^
PE(P-16:0/20:5)	Phosphatidylethanolamine Plasmalogen	−0.95	3.47 × 10^−2^
PE(P-18:0/16:0)		−1.24	6.36 × 10^−3^
PE(P-18:0/20:2)		−1.10	2.16 × 10^−2^
PE(P-18:0/20:5)		−0.94	4.43 × 10^−2^
PE(P-18:0/22:5)		−0.88	3.48 × 10^−2^
PE(P-18:0/22:6)		−0.83	4.23 × 10^−2^
PE(P-18:1/16:0)		−1.34	5.72 × 10^−3^
PE(P-18:1/18:1)		−0.90	2.94 × 10^−2^
PE(P-18:1/20:3)		−1.05	2.28 × 10^−2^
PE(P-18:1/20:4)		−1.01	2.45 × 10^−2^
PE(P-18:1/22:5)		−1.00	3.45 × 10^−2^
PE(P-18:1/22:6)		−1.12	1.10 × 10^−2^
PI(18:0/16:0)	Phosphatidylinositol Ester	−0.82	3.65 × 10^−2^
SM(22:1)	Sphingomyelin	−0.81	3.13 × 10^−2^
SM(24:0)		−1.09	1.68 × 10^−2^
SM(24:1)		−0.95	2.35 × 10^−2^
SM(26:0)		−0.98	2.57 × 10^−2^
SM(26:1)		−1.20	3.52 × 10^−3^
TAG56:3−FA20:2	Triacylglycerol Ester	−1.15	4.07 × 10^−2^
TAG56:4−FA22:4		−1.38	3.19 × 10^−2^
TAG56:5−FA22:5		−1.49	2.66 × 10^−2^
TAG56:6−FA22:6		−1.45	3.16 × 10^−2^
TAG58:10−FA22:5		−1.35	3.27 × 10^−2^
TAG58:10−FA22:6		−1.31	4.34 × 10^−2^
TAG58:6−FA16:0		−1.36	3.69 × 10^−2^
TAG58:6−FA18:0		−1.25	4.19 × 10^−2^
TAG58:6−FA22:4		−1.31	3.42 × 10^−2^
TAG58:6−FA22:5		−1.43	2.79 × 10^−2^
TAG58:7−FA16:0		−1.22	3.33 × 10^−2^
TAG58:7−FA18:0		−1.30	3.26 × 10^−2^
TAG58:7−FA18:1		−1.33	3.09 × 10^−2^
TAG58:7−FA18:2		−1.23	3.79 × 10^−2^
TAG58:7−FA22:4		−1.35	3.49 × 10^−2^
TAG58:7−FA22:5		−1.47	2.35 × 10^−2^
TAG58:7−FA22:6		−1.51	2.94 × 10^−2^
TAG58:8−FA18:1		−1.24	4.39 × 10^−2^
TAG58:8−FA18:2		−1.37	3.27 × 10^−2^
TAG58:8−FA22:5		−1.42	2.63 × 10^−2^
TAG58:8−FA22:6		−1.49	2.72 × 10^−2^
TAG58:9−FA18:2		−1.35	3.24 × 10^−2^
TAG58:9−FA22:5		−1.37	4.12 × 10^−2^
TAG58:9−FA22:6		−1.53	2.74 × 10^−2^
TAG60:10−FA22:5		−1.79	1.11 × 10^−2^
TAG60:10−FA22:6		−1.70	1.33 × 10^−2^
TAG60:11−FA22:5		−1.83	8.01 × 10^−3^
TAG60:11−FA22:6		−1.78	1.14 × 10^−2^
TAG60:12−FA22:6		−1.89	8.91 × 10^−3^

DAG—Diacylglycerols; LPE—Lysophosphatidylethanolamine; MAG—Monoacylglycerol; PC—phosphatidylcholine; PE—Phosphatidylethanolamine; PI—Phosphatidylinositol; SM—Sphingomyelin; TAG—Triacylglycerol; FA—Fatty acid chain.

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
