# Peer review of "Metabolic Signatures of Tumor Responses to Doxorubicin Elucidated by Metabolic Profiling in Ovo"

_metabolites, 2020, doi:10.3390/metabo10070268_

Round 1

Reviewer 1 Report

This work is of high quality with a very clear objective namely the detection of a metabolic signature in ovo. The article is very clear and very well written.

Author Response

We thank the reviewer for this positive appreciation

Reviewer 2 Report

The manuscript titled "Metabolic signatures of tumor responses to

3 doxorubicin elucidated by metabolic profiling in ovo." submitted to Metabolites, describes a very interesting and useful approach using metabolomics in combination with in ovo tumor models to provide a robust platform for drug testing to reveal tumor specific treatment target. According authors knowledge, metabolic profiling has not been applied yet to monitor anticancer drug responses in ovo, therefore work touches an innovative solution. The study seems to be properly designed and authors input a lot of effort to reach valuable results and proper conclusions.

Regarding structure of manuscript and content: Abstract and introduction is well written divided into subsections. Methods and Materials are very well organize. Results and Discussion are presented very clearly and describe all experiments very well. Tables and figures are presented correctly, RSD are calculated and presented as well.

Comments:

Authors didn’t provided QCs in figure 2C and 2D in PCA analysis. Please confirm you check stability of system and reliability of results based on QC and please provide results (QCs cluster on graph)?

Please uniform: PC1/PC2 or PCA1/PCA2 as presently you have different labelling of graphs.

Please provide statistics into PCA graphs: % expl. Var.

p-value: please provide values with comma as it could be more readable.

Funding (line 476-480): please provide necessary information and fill gaps.

Reviewer 3 Report

Dear Editor,

Pleased find my comments of the article by Iman W. Achkar et al entitled “Metabolic signatures of tumor responses to 3 doxorubicin elucidated by metabolic profiling in ovo”.

The manuscript is clearly written and the figures are very good. I think the topic is relevant too.

I only have to minor points for improvement:

  • Figure 1A. The authors must send pictures with higher magnification and better quality. Also, the picture must have scale bar.
  • Figure 1B. It is missing the point a 0 hours, the 100%. Also, the figure is missing the legend
  • Figure 1 D. It is missing scale bar.
  • Figure 1E. It is missing the x axis name.
  • How many eggs were used for tumour growth analysis and how many for metabolomics analysis.
  • Figure legend must be improved. It is missing the doses of doxorubicin treatment in ovo.
  • The authors must perform in the paraformaldehyde fix tumours a Haematoxylin & Eosin staining (tumour morphology), also they must perform Ki67 (to detect cell proliferation), caspase 3 cleavage (detect apoptosis) and human vimentine (human cells). These staining will show if doxorubicin in ovo stop cell proliferation with or without apoptosis induction. This is important for the metabolomics analysis interpretation.
